# Presynaptic endoplasmic reticulum regulates short-term plasticity in hippocampal synapses

Nishant Singh[1], Thomas Bartol[2], Herbert Levine ● [3], Terrence Sejnowski[2] & Suhita Nadkarni ● [1✉]

Short-term plasticity preserves a brief history of synaptic activity that is communicated to the postsynaptic neuron. This is primarily regulated by a calcium signal initiated by voltage dependent calcium channels in the presynaptic terminal. Imaging studies of CA3-CA1 synapses reveal the presence of another source of calcium, the endoplasmic reticulum (ER) in all presynaptic terminals. However, the precise role of the ER in modifying STP remains unexplored. We performed in-silico experiments in synaptic geometries based on reconstructions of the rat CA3-CA1 synapses to investigate the contribution of ER. Our model predicts that presynaptic ER is critical in generating the observed short-term plasticity profile of CA3-CA1 synapses and allows synapses with low release probability to operate more reliably. Blocking the ER lowers facilitation in a manner similar to what has been previously characterized in animal models of Alzheimer's disease and underscores the important role played by presynaptic stores in normal function.

[1] Division of Biology, Indian Institute of Science Education and Research, Pune, Maharashtra, India. [2] Computational Neurobiology Laboratory, Salk Institute for Biological Studies, La Jolla, CA, USA. [3] Center for Theoretical and Biological Physics, Rice University, Houston, TX, USA. ✉email: suhita@iiserpune.ac.in

Synaptic transmission at small central synapses such as CA3–CA1 Schaffer collaterals in the hippocampus is a tightly orchestrated event. At a presynaptic bouton, an action potential triggers the opening of voltage-dependent calcium channels (VDCCs) transiently increasing local calcium concentration. Calcium sensors (synaptotagmins) present on docked vesicles sense the elevated calcium to initiate vesicle exocytosis. The spatial organization of the docked pool of vesicles and VDCCs, and biophysical properties—concentration, binding rates, and diffusion rates—of calcium, VDCCs, calcium sensors, and buffers determine the success of transmission (release probability of neurotransmitter, $P_r$) and play a critical role in constituting its plasticity profile[1].

At the CA3–CA1 presynaptic terminal, the transmission success in response to an action potential is conspicuously low, typically ~10–20%[2]. This apparent flaw is an important feature that allows vesicle release probability to be tunable and the synapse to be plastic. The presynaptic terminal of CA3 pyramidal neurons has a relatively small readily releasable pool (RRP) size of 5–10 vesicles that are available for release. This synapse requires a delicate balance between conserving the vesicle resource and facilitating transmission[3]. A synaptic arrangement of a significant number of VDCCs at a relatively large average distance from the site of vesicle release (active zone) aids this facilitation[1,4,5]. The extended distance attenuates calcium concentration at the active zone (AZ) which ensures a low baseline probability of vesicle release. The large number of VDCCs guarantees abundant bulk calcium that can modulate subsequent releases.

The endoplasmic reticulum is extensively present in the axons of different types of neurons[6–12]. CA3 pyramidal neurons have an ER that is closely associated with the presynaptic terminal[13–16]. In reconstruction studies of rat CA3–CA1 synapses[17] where ~400 boutons were sampled, all boutons, without exception, had a substantial presence of ER; see Supplementary Fig. 1 (K Harris, personal communication). ER (besides functions not relevant for this study) acts as an internal calcium reservoir. ER sequester calcium ions through the action of smooth endoplasmic reticulum calcium transport ATPase (SERCA) pumps, that are released into the cytosol via ryanodine receptors (RyRs) and/or inositol (1, 4, 5)-trisphosphate receptors (IP3Rs). Despite the ubiquitous presence of the endoplasmic reticulum, its role in presynaptic calcium dynamics and concomitant plasticity remains unexplored, due to the experimental difficulty in disambiguating the contributions of different sources of calcium.

Here, we investigate the contributions of ER to calcium signaling and short-term plasticity (STP) in CA3–CA1 synapse. We developed a biophysical model of the CA3 presynaptic terminal reconstructed from EM data[17] (see Fig. 1a, b). The model incorporates essential components of presynaptic calcium dynamics and neurotransmitter release—VDCCs, plasma membrane calcium ATPases (PMCA), calcium buffers (predominantly calbindin in the presynaptic terminal), SERCA pumps, ryanodine receptors, and IP3 receptors[18–23]—organized in a spatially realistic canonical bouton that allows for detailed in silico experiments. We simulated the 2D and 3D diffusion of individual molecules and their reactions using Monte Carlo algorithms in the designated synaptic spatial domain (details in "Methods"). We had previously shown that correctly choosing the model's biophysical properties enables quantitative modeling of characteristic transmitter release the data at this synapse; these data include timescales of release, initial release probabilities, the amplitude of release, etc.[1,2,24–26]. We also simulate different simplified geometries typical of the CA3–CA1 neuropil to show that our results are not limited to specific bouton morphologies (see Fig. 1c, d, e1–3).

## Results

**Synaptic transmission is a tightly choreographed event**. The arrival of an action potential (AP) at an axonal bouton opens VDCCs leading to a large $Ca^{2+}$ influx (Fig. 2a). Mobile calcium buffer calbindin-D28k and SERCA pumps rapidly sequester a large fraction of incoming calcium (see Fig. 2b). Consequently, rapid increase and fall in calcium concentration closely follow the VDCC calcium flux, which, in turn, follows the voltage profile (see Fig. 2c). These buffers govern the calcium signaling in the presynaptic bouton by efficiently limiting the spatial and temporal extent of free calcium in the cytosol. As a result, local calcium concentration at the active zone is high while the bulk calcium in the cytosol remains low. It is this local calcium that is sensed by the docked vesicles at the active zone that leads to a characteristic release probability of ~10–20%. Also, the local calcium concentration despite dropping rapidly does not come back to basal level but takes ~100 ms to slowly reach the base level (Fig. 2c). These lingering calcium ions are involved in the facilitation of release at the next pulse (described in the next section). The PMCA pumps work to restore calcium concentration to its basal level (~100 nM, Fig. 2d). The opening of ryanodine receptors follows the cytosolic calcium profile, whereas IP3 receptors open slowly (Fig. 2e).

Synaptotagmin-1 and synaptotagmin-7—responsible for the synchronous and asynchronous release, respectively—are the major calcium sensors for vesicle release in CA3 pyramidal neurons[27]. In response to an action potential, we see two timescales of vesicle release: immediate vesicle release due to low-affinity and fast calcium-binding sites of synaptotagmin-1 during high calcium concentrations around the active zone; slow release due to high-affinity but slow calcium-binding sites of synaptotagmin-7. Synaptotagmin-1 is the major contributor of the releases as can be seen in Fig. 2f.

**Blocking ER compromises short-term plasticity**. We simulated two different synaptic configurations of the model. We define "Control" synaptic configuration as consisting of RyRs and SERCA pumps on the ER; consistent with extensive reports on presynaptic RyRs[14,28]. However, the expression of presynaptic group I mGluR (types 1 and 5) associated with IP3 production is controversial[29,30]. The other configuration is "Stores Blocked": SERCA pumps are blocked, therefore, no calcium is present in the ER (simulates experimental conditions of using thapsigargin to block SERCA pumps) (Fig. 3a).

We used the paired-pulse ratio (PPR), a classic measure of presynaptic STP to characterize plasticity profiles. In a PPR protocol, axons are stimulated by two spikes separated by an interspike (time) interval (ISI). Here, we defined PPR as the ratio of the probability of neurotransmitter release due to the second stimulus ($Pr_2$) to that of the first stimulus ($Pr_1$), averaged over multiple trials[2,31]. The inverse relationship observed between the release probability and the PPR is a universal feature for small hippocampal synapses[2,26,31]. One reason for this is that the release probability is bounded by 1. This limits the subsequent increase in $Pr_2$ for synapses with preexisting high $Pr_1$ and thus limits the increase in PPR. More importantly, for synapses with very high intrinsic release probabilities, depletion of the small RRP overwhelms calcium-driven facilitation and gives rise to low PPR. Synapses with large numbers of VDCCs have a higher intrinsic release probability as a consequence of a large calcium signal and therefore, operate at a low PPR. Although most CA3 terminals operate at a characteristic low release probability and as a result can display a large PPR. This PPR is tunable in an activity-dependent manner[32] and is a fundamental component of STP at this synapse.

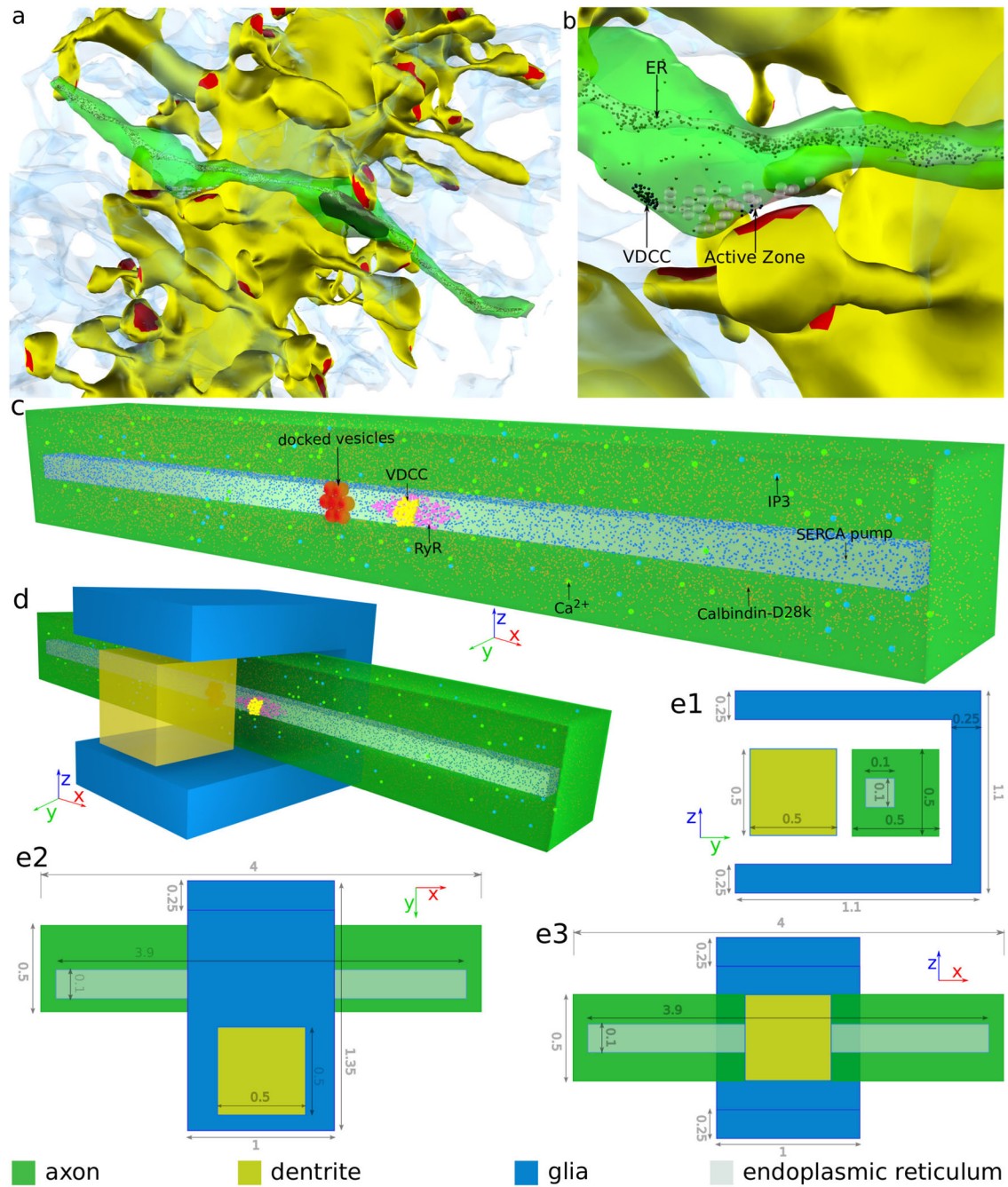

**Fig. 1 Model geometry. a** Reconstruction of CA3–CA1 neuropil in rat hippocampus. The dendrite of a CA1 pyramidal neuron is shown in yellow with red patches, indicating the postsynaptic densities. An axon of CA3 pyramidal neuron is shown in green (other axons in the vicinity have been made transparent for clarity). ER inside the axon is shown in gray. Astrocytes are shown as translucent blue structures. **b** An en passant synapse formed by a CA3 axon onto a CA1 dendrite. **c** Presynaptic terminal of the 3D canonical model showing the placement of key molecules. **d** The complete view of the canonical model showing the relative arrangement of glia, pre- and postsynaptic terminals. **e1–3** Dimensions (in µm) of the model in orthographic projection. ER endoplasmic reticulum, VDCC voltage-dependent calcium channel, RyR ryanodine receptor, IP3 inositol (1,4,5)-trisphosphate.

In the Control synapses, the presence of SERCA reduces the total free calcium. SERCA pumps, characterized by high-affinity and fast-binding sites for calcium[33–35], can take-up a substantial amount of calcium arriving through VDCCs. In contrast, increased free calcium in the Stores blocked synapses leads to an overall increase in release probability compared to Control synapses for the same calcium influx. This is described in Fig. 3b1 for reconstructed synapse and Fig. 3b2 for a synapse with simplified geometry with the average size of a CA3 presynaptic terminal that we call a canonical synapse (see Fig. 1c–e). Blocking

store activity accordingly shows reduced PPR at any given release probability compared to the Control synapse (red, Fig. 3c1 for reconstructed synapse and Fig. 3c2 for canonical synapse). This effect is most prominent in the low release probability regime where most CA3–CA1 synapses operate. Rapid binding and slow unbinding of calcium by SERCA pumps in response to two pulses separated by 40 ms are shown in Fig. 3d). As a consequence, the peak amplitude of calcium signal measured at the active zone, for the Control synapse, is lower than it would be with Stores blocked. The lower calcium peak that maintains a lower release

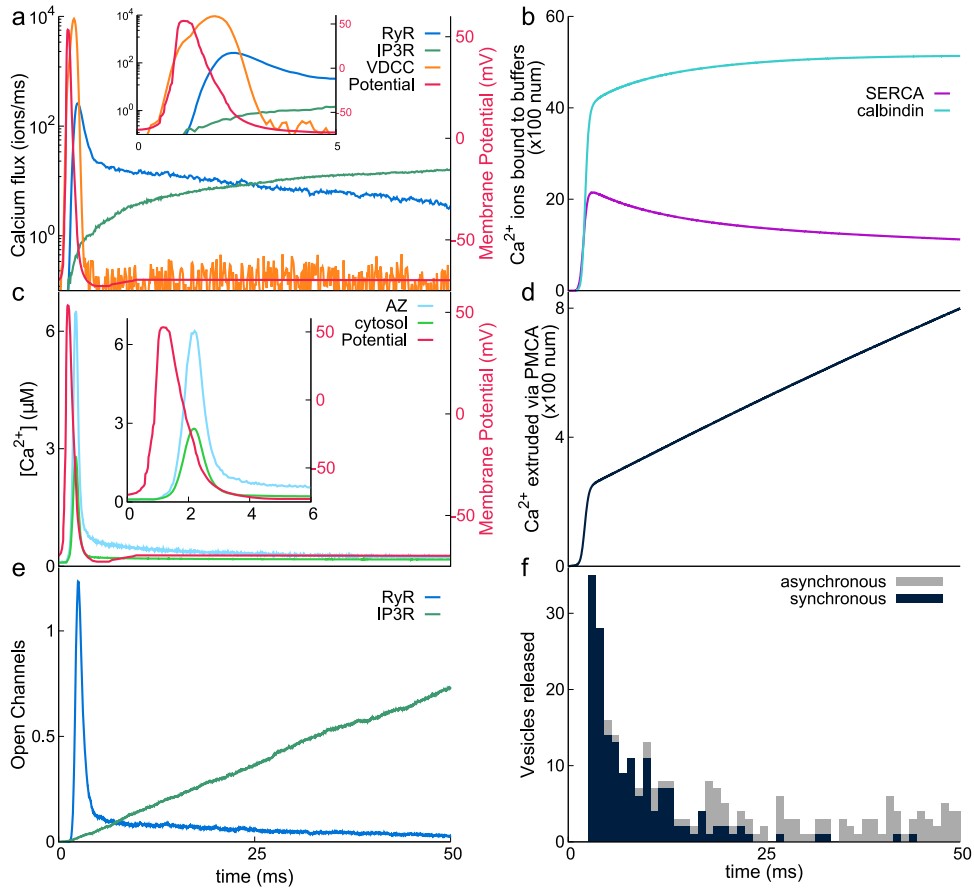

**Fig. 2 Neurotransmission in response to a single AP arriving at the synapse for 2000 trials. a** Depolarization due to AP (red, right "*Y*" axis) leads to an influx of $Ca^{2+}$ ions via VDCC (orange) and subsequently via RyR and IP3R (blue and green, respectively). Inset shows the first 5 ms of the simulation. **b** Calbindin-D28k (cyan) and SERCA pumps (pink) are seen to buffer a large part of $Ca^{2+}$ influx from VDCC. **c** Calcium concentration at the active zone, $[Ca^{2+}]_{AZ}$ (light blue) measured 10 nm above the active zone and in the entire presynaptic bouton, $[Ca^{2+}]_{cyt}$ (light green). Inset shows the first 6 ms of the simulation. **d** PMCAs actively pump $Ca^{2+}$ ions from the cytosol to maintain a base level $[Ca^{2+}]$ of 100 nM. **e** Activity of RyR (blue) and IP3R (orange) show distinct responses to an AP. **f** Stacked histogram of vesicle fusion (neurotransmitter release) events via synchronous (black) and asynchronous (gray) pathways in response to a single AP for 2000 trials. There were a total of 269 vesicle releases in 194 trials—some trials had multiple releases—(out of 2000) within the 50 ms of the simulation. The average release probability, in this case, is 194/2000 (fraction of successful trials).

probability for Control synaptic configurations is seen consistently across a wide range of VDCC expressions (Fig. 3e) and is evident from the vesicle release profile (see Supplementary Fig. 2). Interestingly, subsequent unbinding of calcium from the SERCA (Fig. 3d) results in larger cumulative calcium at the active zone in Control synapses (Fig. 3f). Longer-lasting residual calcium (Fig. 3g) in the Control synapse facilitates vesicle release upon subsequent stimulation. The enhanced PPR for Control synapses occurs over a wide range of ISI (Fig. 3h1–2). The effect of enhanced PPR is stronger for shorter ISI and corresponds to the residual calcium decay in the synapse (see Supplementary Fig. 3 for simulations with an ISI of 20 ms). In summary, the facilitation orchestrated by SERCA pumps is robust across a wide range of intrinsic release probabilities and ISI.

These results reveal an important, previously unaccounted role for SERCA in determining short-term plasticity. These results are not sensitive to reasonable variations in SERCA densities and rates of SERCA binding (see Supplementary Fig. 4) or synaptic geometries (see Fig. 3c1, c2). Blocking RyR in the Control synapse does not change *Pr* since RyR does not contribute to the calcium signal sufficiently (Supplementary Fig. 5). Despite the close juxtaposition of RyR and VDCCs (50 nm), calcium release via RyRs opening is not a major contributor to timescales relevant to PPR shown here.

**Contribution of intracellular stores to facilitation in response to a stimulus train.** Next, we investigated facilitation (defined as the ratio of vesicle release probability due to the n^th stimulus to the 1^st stimulus) in synapses when stimulated with a train of action potentials (20 APs at 20 Hz, Fig. 4; see Supplementary Figs. 6 and 7 for stimuli at 10 Hz and 50 Hz, respectively. As in the previous section, we simulated two different synaptic configurations of the model: "Control" and "Stores Blocked" in both reconstructed and canonical geometries.

Both synaptic configurations, Control and Stores blocked, show a peak in facilitation that decays with subsequent stimuli (Fig. 4a1–2). Similarly, a peak in release probability is also seen (Fig. 4b). Parallel to the PPR protocol, a high baseline release probability corresponding to a synapse with Stores blocked (red, 4b1–2) is linked to low facilitation (red, Fig. 4a1–2). The initial increase in vesicle release probability with stimulus number leads to a rapid depletion of the RRP. The decrease in release probability after the peak is dictated by the depletion of RRP (Fig. 4c) and is evident from the vesicle release profile (see Supplementary Fig. 8). This competition between the increase in release probability and the subsequent decrease in the availability of vesicle resources (seven vesicles in RRP) lowers facilitation after the initial increase for all synaptic configurations. As in the previous protocol, SERCA pumps rapidly bind a significant

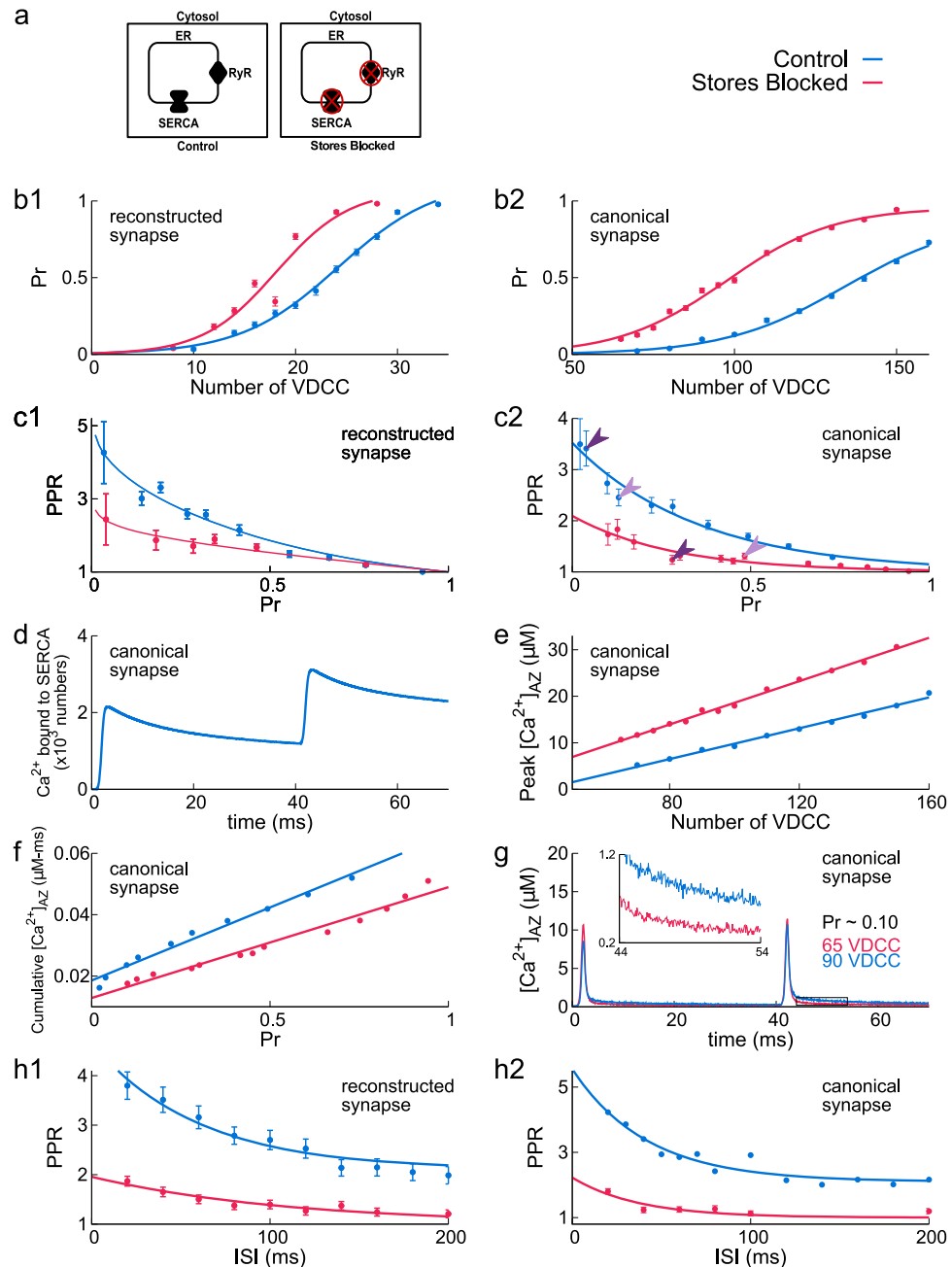

**Fig. 3 ER contribution to paired-pulse ratio for 40 ms ISI. a** Two model setups—"Control": ryanodine receptors and SERCA pumps are present on the ER; "Stores Blocked": SERCA pumps are blocked, therefore, no calcium ion is present in ER and RyR is redundant. **b1–2** Variation of release probability of a vesicle (Pr) with number of VDCCs. **c1–2** Inverse relation of paired-pulse ratio (PPR) and intrinsic Pr for various synaptic configurations. Dark purple arrows indicate the PPR corresponding to 80 VDCCs, and light purple corresponds to 100 VDCCs for canonical synapse. **d** Amount of $Ca^{2+}$ ions bound to SERCA in response to a paired pulse. **e** Variation in peak calcium concentration at the active zone, $[Ca^{2+}]_{AZ}$, with the number of VDCCs. **f** Cumulative calcium concentrations over 20 ms at the active zone in response to the second AP in a paired-pulse protocol for various Pr. **g** Calcium concentration at the active zone, $[Ca^{2+}]_{AZ}$ for $Pr = 0.10$. The colored text describes the corresponding number of VDCCs in each of the configurations to arrive at $Pr = 0.10$. Inset: box area zoomed-in to show details of base level $[Ca^{2+}]_{AZ}$ concentration after the second AP. **h1–2** Variation of paired-pulse ratio for different ISI. Data are mean ± s.d.

number of incoming calcium during an action potential (Fig. 4d, top) in addition to calcium binding by calbindin-D28k (Fig. 4d, bottom). The Control synapse has lower peak calcium at the active zone (Fig. 4e1). Bulk calcium also shows a similar trend (Fig. 4f1). However, notice that residual calcium concentrations (post AP) are higher (Fig. 4e2, f2) for the Control synapse. As in the previous case, calcium buffering by SERCA maintains a lower Pr and therefore, high facilitation; until vesicle depletion becomes

more important. IP3R and RyR have minor contributions to this form of facilitation and have also been investigated in detail (see Supplementary Fig. 9).

**Increased reliability mediated by stores machinery.** Experimental data on spatial navigation and sensory processing suggest that information is encoded in small hippocampal synapses via

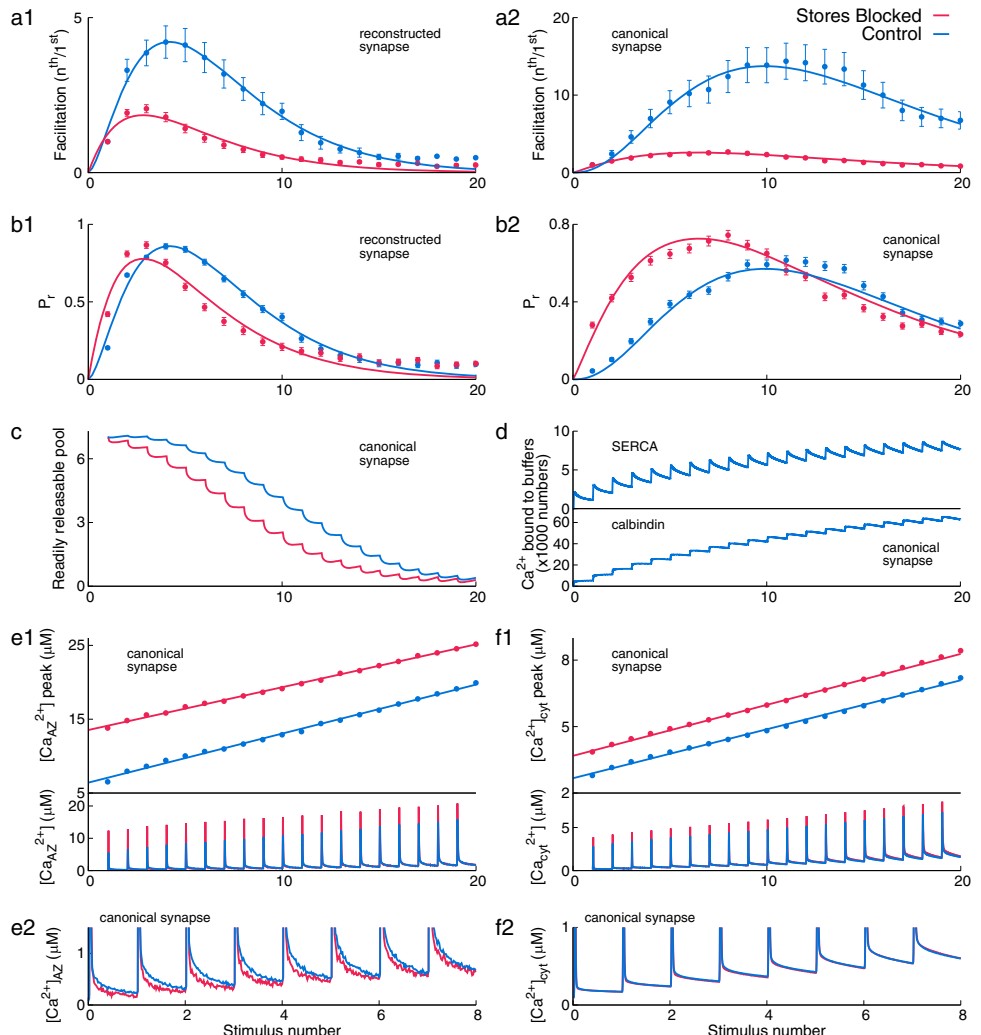

**Fig. 4 Response of the bouton to a stimulus of 20 pulses at 20 Hz. a1–2** Facilitation for a synapse with Stores blocked is lower than Control. **b1–2** Release probability of a vesicle (Pr) for each AP in the train. **c** Decrease in the RRP due to successive releases. **d** Calcium ions binding to SERCA (top) and calbindin-d28k (bottom). **e1** Peak of calcium at the active zone, $[Ca^{2+}]_{AZ}$ (top) and the trace of average calcium at the active zone (bottom), and (**e2**) zoom-in of the bottom panel of (**e1**) showing the differences in basal calcium levels. **f1** Peak of calcium in the entire bouton, $[Ca^{2+}]_{cyt}$ (top) and the trace of average calcium in the entire bouton (bottom), and (**f2**) zoom-in of the bottom panel of (**f1**) showing the differences in basal calcium levels. Data are mean ± s.d.

finely tuned activity rates[36,37]. This is achieved despite the highly stochastic nature of individual synapses operating in the low release probability range. Here, we show that intracellular calcium stores and the associated machinery may reduce the pulse-to-pulse variability of synapses with low release probabilities of vesicles. For a facilitating synapse, $Pr_2$ increases with $Pr_1$ (right "$y$" axis in Fig. 5a) and as seen before, synapses with calcium stores (Control) facilitate more strongly compared to synapses with Stores blocked. A synapse with stores displays the higher success of transmission of consecutive spikes is an obvious implication of enhanced facilitation seen in these synapses. Another way to characterize a more reliable rate code arising out of enhanced vesicle release probability to subsequent releases (facilitation) is coefficient of variance (CV), defined as the standard deviation of $Pr_2$/mean $Pr_2$. CV remains lower for Control synapses compared to Stores blocked as shown in left "$y$" axis, solid lines in Fig. 5a indicating a less fluctuating release rate. As expected, CV of $Pr_1$ follows $((1 - Pr)/n \times Pr)^{1/2}$ (for a binomial distribution; $n$ is the number of active zones; see Supplementary Fig. 10).

So far we have described features of a population of presynaptic terminals in the Schaffer collaterals rather than the relevance of activity as seen by a single synapse. It has been suggested that crucial hippocampal functions require rapid "synapse-specific" changes that guide more gradual changes before being consolidated in neocortical networks for permanent storage[38–41]. To characterize the "synapse-specific" history of activity applicable for STP timescales, rather than statistics of independent releases, we next focus on the conditional probability, $P_{11}$[42,43]. $P_{11}$ is defined as the probability that a successful transmission event in response to a first stimulus is followed by another successful transmission event in response to the second stimulus.

The negative influence of vesicle depletion on $P_{11}$ due to the previous release is identical for both types of synapses with the same size of RRP. However, $P_{11}$ is also influenced by the complex, noisy, nonbinary presynaptic calcium signal that may persist after the stimulus. Conditional probability, $P_{11}$ is higher for synapses with calcium stores, across all $Pr_1$, and is shown in Fig. 5b. $P_{11}$ is also higher for a wide range of ISIs for a synapse with stores (see Fig. 5c). This indicates that a synapse with stores retains a better history of release and calcium dynamics. It has been shown that efferent synapses are involved in critical computations and short-term plasticity enables estimating nondiscrete signals arriving at the synapse[44]. Our analysis suggests that a synapse with a higher

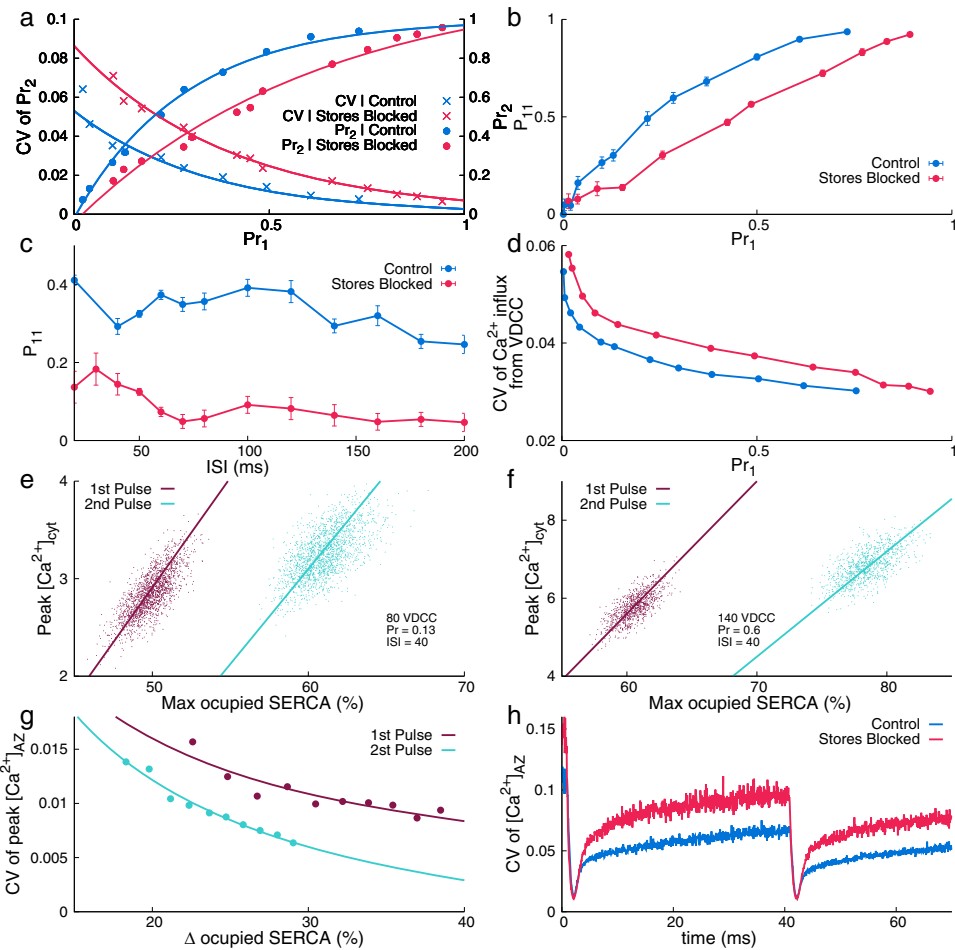

**Fig. 5 Reliability of synapses with ER. a** Release probability of a vesicle in response to the second pulse ($Pr_2$, right "y" axis, fitted with dotted lines) in a paired-pulse protocol and coefficient of variation of $Pr_2$ (left "y" axis, fitted with solid lines) as a function of release probability in response to the first pulse ($Pr_1$). **b** Conditional probability ($P_{11}$) as a function of $Pr_1$. **c** $P_{11}$ for a range of interspike interval, ISI (20–200 ms) with an intrinsic release probability, $Pr \sim 0.14$. **d** CV of calcium flux through VDCCs as a function of $Pr_1$. The larger number of VDCCs contribute to release in Control synapses (see Fig. 3a) resulting in a lower CV of calcium flux. **e**, **f** Correlation between peak cytosolic calcium concentration ($[Ca^{2+}]_{cyt}$) and the % maximum occupancy of SERCA pumps for synapses with intrinsic $Pr \sim 0.13$ and $Pr \sim 0.6$ (2000 and 1000 trials, respectively). **g** Decreasing trend of CV of the peak of calcium concentration at the active zone, $[Ca^{2+}]_{AZ}$, as a function of %Δ occupied SERCA sites. %Δ occupied SERCA = ((max occupancy − initial occupancy) × 100/total SERCA-binding sites). **h** CV of $[Ca^{2+}]_{AZ}$ across 2000 trials for a synaptic configuration with 80 VDCCs. Data are mean ± s.d.

$P_{11}$ as seen with calcium stores can better capture this nondiscrete nature of information (calcium signal) arriving at the synapse. Thus, the presence of intracellular calcium stores and the residual calcium ensures a noisy synapse with a prescribed intrinsic property (such as a low $Pr$) is a better predictor of ensuing release.

At fixed $Pr_1$, a larger number of VDCCs are needed in synapses with stores to arrive at a release threshold (see Fig. 3c), and hence, the coefficient of variation (CV) of the calcium signal is lower, regulated by a larger number of VDCCs (Fig. 5d). A synapse with stores clearly shows enhanced calcium leftover from previous activity (Fig. 3g). Both (larger residual calcium and low CV of VDCCs) contribute to the enhanced facilitation manifested in $P_{11}$ for Control synapses. Since a lower CV of calcium signal through VDCCs in Control implies that a larger than "average" signal that caused the first success is consistently larger across trials and therefore more reliably causes the second success in the release ($P_{11}$). In addition to the role played by VDCCs, calcium signals are less noisy for Control, due to the role played by SERCAs. SERCA binding keeps up with the noise in calcium peaks across trials. This is shown as a strong correlation between maximal % of SERCA-binding sites and peak calcium during a paired-pulse stimulus in Fig. 5e for both (low $Pr_1$) and 5f (high $Pr_1$). We also

show that the CV of peak calcium concentration decays with an increase in SERCA occupancy as intrinsic release probability ($Pr_1$) is varied (see Fig. 5g). The "x axis" or each Δ occupied SERCA (%) corresponds to a specific synaptic release probability configuration (incoming calcium corresponds to intrinsic release probability and therefore change in SERCA occupancy). Thus SERCAs smoothens calcium fluctuations for the entire span of the intrinsic release probability range (see Fig. 5g). The isolated influence of SERCA without the contribution of voltage-gated calcium channels in lowering the variability of calcium is described in Fig. 5h. Essentially, we simulated synapses with an identical number of VDCCs for both Control and Stores-blocked. The effect of SERCA has clearly seen between the pulses wherein the CV of calcium signal at the AZ remains lower for Control synapses. In summary, intracellular calcium stores allow synapses to operate at low intrinsic release probability and yet exhibit large, facilitated release with a lower CV caused by longer-lasting and less noisy residual calcium.

That the CV of calcium response at the active zone for Control is lower compared to a synapse with Stores blocked (Fig. 5h) is yet another independent observation. The less noisy calcium signal associated with the Control synapse may also have implications in

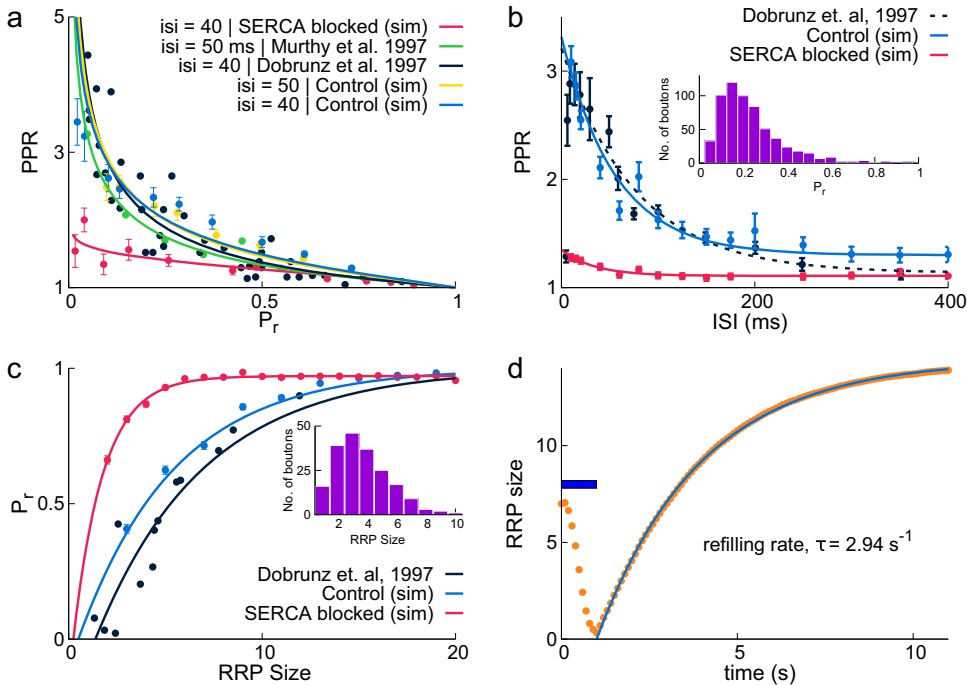

**Fig. 6 Comparison of simulation results with experimental data. a** Inverse relation of paired-pulse ratio (PPR) with intrinsic release probability (Pr) for 40 ms ISI (experiment, black; simulation, blue) and 50 ms ISI (experiment, green; simulation, yellow), and Stores blocked (simulation, red) (experimental data from Dobrunz and Stevens[25,26]). **b** PPR decreases with increasing ISI for all synapses: Control (blue), Stores blocked (red), and experiment (dashed, black). Inset shows the distribution of Pr in synapses used in the experimental result (data from Murthy et al.[26]) and in simulations. **c** Variation of Pr with readily releasable pool (RRP) size for Control (blue), Stores blocked (red), and experimental data (black). The distribution of the RRP size used for simulations is shown in inset[81]. **d** RRP (shown in orange) was depleted by a 20 Hz stimulus for 1 s (indicated by horizontal blue bar) and vesicle refilling was observe and fitted to an exponential curve (blue) (replenishment timescale $\tau = 2.94$ s). Data are mean ± s.d.

other calcium-dependent downstream signaling involving vesicle recycling[45].

**Intracellular calcium store necessary for PPR requirement of a CA3–CA1 synapse.** Thus far we described the mechanisms by which SERCA and other components associated with calcium release from ER influence synaptic transmission. Here, we compare the predictions of our model with experimental observations of short-term plasticity at CA1–CA3 synapses[2,26].

In Fig. 6a, we show PPR for the entire range of intrinsic release probabilities as predicted by the model (Control and with Stores blocked) and as measured in two independent experiments. Simulations of Control synapses that include presynaptic calcium stores are in better agreement with experimental data for both 40 and 50 ms ISI[2,26]. Figure 6b shows PPR as a function of ISI. Our simulations lead us to expect that as the ISI increases, the probability of influencing the second release by residual calcium decreases. This is consistent with experiments by Dobrunz and Stevens[25] for CA3–CA1 synapses. In order to replicate these experiments, we incorporate distribution of release probabilities as seen in CA3 presynaptic terminals as described by Murthy et al.[26]. The PPR response of the model closely followed the experimental measurements for the Control synapse, in which stores were included. In contrast, when SERCA was blocked, the PPR response was reduced drastically and was not in agreement with experimental data.

The dependence of release probability on RRP size was measured in a different set of experiments by Dobrunz et al.[2]. These experiments were carried out with higher extracellular $Ca^{2+}$ concentration (4 mM) to minimize the amount of facilitation; our model calculations incorporated higher extracellular calcium concentration (4 mM) for both the Control

synapse and the synapse with Stores blocked (see Fig. 6c). A better match with experiments was found for the Control synapse. When SERCAs were blocked, we found a much higher vesicle release probability, which was saturated to almost 1 for a very small number of vesicles (~5). This higher $Pr_1$ response in simulations with calcium Stores blocked can again be attributed to the absence of buffering by SERCA. Finally, the timescales for vesicle recycling can critically modify both PPR and release probabilities. The vesicle recovery after depletion of our model (see Fig. 6d) is in agreement with experimental observations (replenishment timescale: $\tau_{model} = 2.94$ s, $\tau_{experiment} = 2.8$ s[2]).

## Discussion

The presence of ER is reported in both axonal and dendritic compartments[14,46]. Only ~20% of CA1 spines have ER and it is overrepresented in the larger, stronger synapses[47]. In contrast, the ER in CA3 axons seems to extend through all boutons; see Supplementary Fig. 1 (K Harris, personal communication, June 2008). The endoplasmic reticulum can modulate several downstream signaling cascades. Here, we report on the distinct role of presynaptic ER seen extensively in the Shaffer collaterals, in making a crucial contribution to short-term plasticity. This is predominantly via the buffering action of SERCA pumps on the ER membrane. For train stimulus lasting a few seconds, calcium release from ER receptors also contributes to facilitation.

Fast action and high-affinity binding are characteristic features of SERCA[33] and are essential for physiological refilling of the ER. Consistent with these requirements, structural analysis shows that the SERCA2b isoform most prevalent in the hippocampus[34,35] has the highest $Ca^{2+}$ affinity due to a unique C-terminal extension[48]. These intrinsic biophysical properties of SERCA make it highly effective at lowering bulk calcium. SERCA's calcium

buffering properties in turn lower *Pr*. The inverse relationship between release probability and PPR at the CA3 presynaptic terminals is well-established in experiments[2,49]. Calcium from consecutive pulses binds to the open SERCA sites. This decrease in the overall availability of SERCA also decreases its buffering capacity resulting in increased free calcium and increased facilitation. Therefore, a low release probability for synapses with ER ensures high facilitation compared to synapses with no ER. The actual contribution of calcium release from RyR and IP3R over these shorter timescales is minor compared to train stimulus.

At the CA3 terminal, there is a trade-off between short-term facilitation and reliability. Spatial location and sensory information is encoded in the firing rates of pyramidal cells in the hippocampus. Lesions to the hippocampus that impair rate coding can compromise sensory discrimination[36]. Reliable firing rates can be maintained, despite the stochasticity of vesicle release if a large number of synapses are activated with bursts of spikes rather than isolated spikes. However, it has also been argued that the hippocampal rate code describing a location in an environment is represented by the activity of a small number of neurons (population rate code)[50,51]. The alternative strategy would be to operate at high vesicular release probability, but at high *Pr*, the small RRP size at this synapse would deplete quickly. The ER allows synapses to have low intrinsic release probability but still exhibit relatively higher reliability in an activity-dependent manner.

Both IP3R[29,30] and RyR expression have been reported in presynaptic terminals; however, the presence of presynaptic RyR is more prevalent[14,20,28,52–54]. Binding of calcium alone triggers the opening of ryanodine receptors, leading to calcium release. The opening of IP3 receptors, however, requires both IP3 and calcium to be bound. Glutamate released from vesicles can lead to IP3 production via mGluR and G-protein pathways (see "Methods" for details) and lead to IP3R opening and consequential calcium release. Ryanodine receptors have low affinity and require a large calcium transient to open and hence directly follow the VDCC $Ca^{2+}$ flux. IP3 receptors have higher affinity and remain open for several seconds past the VDCC flux termination. The distinct biophysical properties for each of these receptors provide a wide spatiotemporal range for calcium signaling that can be sustained for seconds and participate in several different forms of plasticity. Given the differential expression of these receptors, we have investigated various synaptic configurations and the effect of each of these receptors. We conclude that the presence of presynaptic ER allows a repertoire of multiple timescales for calcium signaling, and for other downstream molecular signaling[55,56].

Our approach has been to build a prototype of the CA3 terminal to carry out in silico experiments and make quantitative (not merely qualitative) predictions. This is especially valuable since direct measurements of crucial molecular signals, such as local calcium at the active zone in this small but important synapse, is difficult. The model accounts for the observed rise time, the decay time of the peak of vesicle release, calcium affinities of the calcium sensors for vesicle release, effective calcium diffusion, the timescale of calcium-transient decay, the ER refilling rate, and steady-state value of ER concentration[24,57,58]. The reported timescale of calcium-transient decay, the ER refilling rate, and the steady-state value of ER concentration constrain the binding rates and expression levels of SERCA in our model[57,58]. PPR is a high-dimensional measure of plasticity and is determined by several interdependent synaptic components (residual calcium, RRP size, buffering capacity, calcium sensors among others). We show that the canonical model that includes intracellular calcium stores (without adjusting for any parameter), agrees well with several independent studies of the CA3

presynaptic terminal. It is reasonable to assume that a biophysical model that quantitatively agrees with the measurement of PPR for a wide range of protocols is physiologically realistic. Several studies have shown that a biological system can achieve a specific goal using multiple sets of parameters[59,60]. Given the profuse degeneracy of biological systems, our aim was not to arrive at an idealized set of parameters. By demonstrating that our model makes a number of nontrivial "post-dictions"[61], we argue that our main prediction on the role played ER in short-term plasticity is also likely to be accurate.

We have implemented loose coupling between calcium channels and the calcium sensor that governs vesicle release. Our model geometry is based on our previous work[1] and independent observations in pyramidal cell axon terminals consistent with Ohana and Sakmann[62] and more specifically in CA3 pyramidal neurons[4,5]. We do not eliminate the possibility of a fraction of channels co-localized at the active zone. Our previous results show facilitation increases with increasing distance between the VDCCs and the AZ. However the PPR profiles, despite implementing a possible upper limit of coupling distance, fall short of the observed values. This suggests the involvement of an additional mechanism in generating physiological facilitation. In Supplementary Fig. 11, we show that our results are robust to the heterogeneous distribution of channels such that a few channels are juxtaposed with the active zone. When a subpopulation of calcium channels are expressed at an extended distance, they contribute minimally to basal release but increase bulk calcium substantially and cause facilitated subsequent releases. The extended geometrical arrangement between VDCCs and the active zone may seem at apparent odds with Holderith et al.[63]. In their data CA2.1 expression, one of the three main contributors to the VDCC signal is primarily seen close to the active zone. However, their results do not eliminate the expression of the other two types, Cav2.2 and Cav2.3 channels that contribute to at least 50% of calcium response, at extended distances. Also, our model captures the effective distance of 350 nm between the calcium channels and docked vesicles in the active zone (measured as average distance) for a wide range of heterogeneous arrangements between calcium channels and the active zone with the same effective distance. We have systematically carried out simulations that verify its equivalence and are shown in Supplementary Fig. 11.

The functional implications of various kinds of short-term plasticity remain critical questions in neuroscience. Short-term plasticity modulates the functional efficacy of synaptic transmission at a millisecond-to-seconds timescale. In the mammalian brain, short-term synaptic plasticity influences the information processing function of synapses, enabling them to optimize network-level computation[44]. Synapses with a low initial probability of release are capable of facilitation and function as high-pass filters, and the reverse is true for synapses with a high initial probability of release[64–66]. Other reported functions of short-term plasticity include insulating postsynaptic terminal from direct stream of presynaptic activity, spike sequence decorrelation, working memory, and, optimizing energy consumption and information transmission[67–71].

Abnormal STP is among the earliest indications of various forms of dementia[72]. In a presenilin animal model of Alzheimer's disease (AD), facilitation in hippocampal CA3–CA1 synapses was reduced compared to Control synapses. This modified facilitation in AD synapses was observed across a range of ISIs for a paired-pulse protocol and a train stimulus protocol. Immediately relevant to the present study, blocking presynaptic ER in normal synapses resulted in a substantial reduction in facilitation[53,54]. Interestingly, this reduction was quantitatively identical to compromised facilitation in AD synapses. Furthermore, the effect of

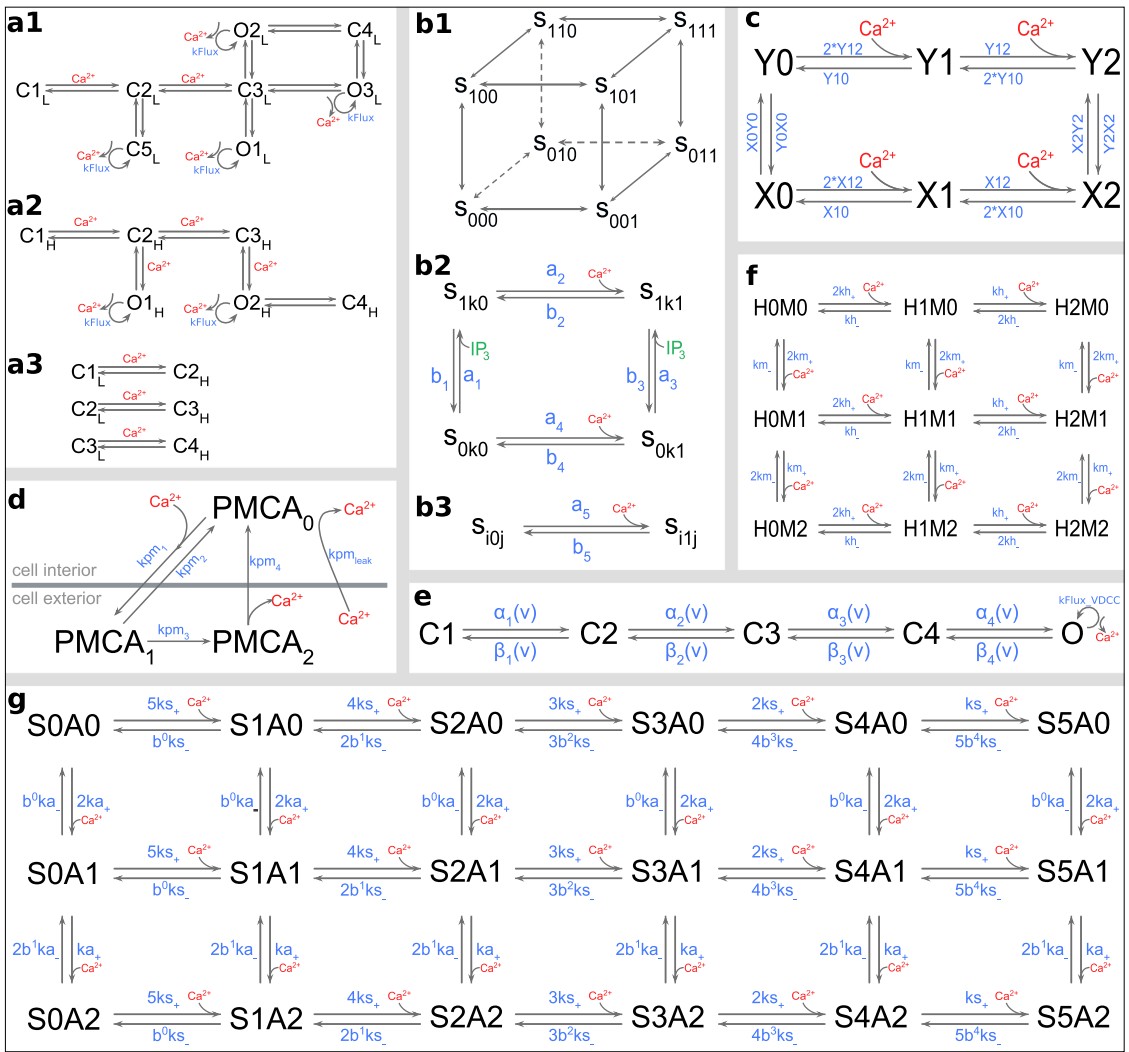

**Fig. 7 Kinetic Schemes. a1–3** RyR-L mode, RyR-H mode, and transition between "L" and "H" modes of RyR. **b1–3** IP3 receptor. Lines connecting a state of IP3 receptor in panel **b1** represents the possible states it can transition to. Corresponding transition rates are mentioned in panel **b2** and **b3**, where **i**, **j**, and **k** can take values 0 and 1. $S_{110}$ is the open state. **c** SERCA pumps. The binding sites described by "X" is on the cytosolic side, while "Y" is on the ER side. Calcium is transported across the membrane via transition between "X2" and "Y2". **d** PMCA pump. The scheme also incorporates the calcium leakage across the plasma membrane. **e** High-threshold voltage-dependent calcium channel. "O" is the open state that conducts calcium. **f** Calcium buffer, calbindin-D28k. It is modeled to have two calcium-binding units, termed as, high "H" and medium "M", each of which binds two calcium ions. **g** Calcium sensor for vesicle release with two subunits, "S" for fast synchronous release with 5 calcium-binding sites and "A" for slow asynchronous release with two calcium-binding sites. The release can take place via either the synchronous or the asynchronous unit. Reaction rates are described in Table 1.

ER blocking was occluded in AD synapses. These experiments indicate that disrupted ER signaling underlies compromised plasticity in AD. These data support our prediction on the role of ER in short-term plasticity and its critical contribution to normal function. Our model does not describe the contribution of presynaptic ER to long-term potentiation (LTP) and the long-term costs of loss of STP as seen in AD, as these are beyond the scope of this present study. We do note that low $Pr$ and high PPR in CA3 boutons correspond to a higher magnitude of LTP[40] and potentially describes a relationship between STP and LTP. We hope to study these issues in future extensions of the current work.

## Methods

Monte Carlo simulations were carried out using MCell (https://mcell.org), version 3.2. MCell simulates the diffusion of individual molecules (with specific diffusion constants) present either on a surface or in a confined volume and carries out user-specified molecular reactions stochastically. We simulated an en passant axon segment with physiologic spatial distributions and concentrations of relevant

molecules. These simulations track each molecule and the relevant reactions to calculate spatiotemporal trajectories. Simulations were performed on a high-performance computing cluster (HP PROLIANT SL230s Gen8 as compute nodes, each with two CPUs containing ten cores each; CPU: Intel(R) Xeon(R) CPU E5-2860 v2 α 2.80 GHz) with 1464 processing units housed in IISER Pune.

**Model components and geometry.** Simulations were carried out in (1) a presynaptic terminal with canonical dimensions ($0.5 \times 0.5 \times 4 \, \mu m$ volume, surface area: $8.5 \, \mu m^2$) representative of a canonical CA3 presynaptic terminal with simplified cuboidal geometry (see Fig. 1c–e), in (2) a CA3 axon reconstructed from EM data with volume $0.39 \, \mu m^3$ (see Fig. 1a, b). The details of reconstruction of neuropil can be found in Harris et al.[17], and (3) a presynaptic terminal with simplified cuboidal dimensions similar to the reconstructed synapse from EM images. The canonical model is composed of three major geometrical components: cuboidal presynaptic, postsynaptic terminal, and a U-shaped astrocyte surrounding the synapse. The presynaptic terminal contains a cuboidal ER compartment of dimensions ($0.1 \times 0.1 \times 3.9 \, \mu m$). We sampled distances between ER fragments and the cytoplasm in 75 randomly selected presynaptic terminals from the reconstructed dataset. The most probable distance was measured to be ~120 nm.

Molecular components of the model and their placement are shown in Fig. 1d. Kinetic schemes and the corresponding reaction rates for the following

**Table 1 Reaction rates for the kinetic schemes used in the simulation.**

| Ryanodine receptor | Saftenku et al.[73] |
|---|---|
| $C1_L - C2_L, C2_L - C1_L$ | $1.24 \times 10^6 \, M^{-1} s^{-1}$, $13.6 \, s^{-1}$ |
| $C2_L - C3_L, C3_L - C2_L$ | $2.98 \times 10^7 \, M^{-1} s^{-1}$, $3867 \, s^{-1}$ |
| $C2_L - C5_L, C5_L - C2_L$ | $1.81$, $3.63 \, s^{-1}$ |
| $C3_L - O1_L, O1_L - C3_L$ | $731.2$, $4183 \, s^{-1}$ |
| $C3_L - O2_L, O2_L - C3_L$ | $24.5$, $156.5 \, s^{-1}$ |
| $C3_L - O3_L, O3_L - C3_L$ | $8.5$, $111.7 \, s^{-1}$ |
| $C4_L - O2_L, O2_L - C4_L$ | $415.3$, $1995 \, s^{-1}$ |
| $C4_L - O3_L, O3_L - C4_L$ | $43.3$, $253.3 \, s^{-1}$ |
| $C1_H - C2_H, C2_H - C1_H$ | $3.26 \times 10^6 \, M^{-1} s^{-1}$, $116 \, s^{-1}$ |
| $C2_H - C3_H, C3_H - C2_H$ | $6.6 \times 10^5 \, M^{-1} s^{-1}$, $163 \, s^{-1}$ |
| $C2_H - O1_H, O1_H - C2_H$ | $7.86 \times 10^6 \, M^{-1} s^{-1}$, $1480 \, s^{-1}$ |
| $C3_H - O2_H, O2_H - C3_H$ | $7.77 \times 10^6 \, M^{-1} s^{-1}$, $330 \, s^{-1}$ |
| $C4_H - O2_H, O2_H - C4_H$ | $2390$, $298 \, s^{-1}$ |
| $C1_L - C2_H, C2_H - C1_L$ | $6.6 \times 10^2$, $0.083 \, s^{-1}$ |
| $C2_L - C3_H, C3_H - C2_L$ | $6.6 \times 10^2$, $0.083 \, s^{-1}$ |
| $C3_L - C4_H, C4_H - C3_L$ | $6.6 \times 10^2$, $0.083 \, s^{-1}$ |
| kFlux | $1.09 \times 10^9 \, M^{-1} s^{-1}$ |
| IP3 receptor | De Young and Keizer[74] |
| a1, a2, a3, a4, a5 | $4 \times 10^8, 2 \times 10^5, 4 \times 10^8, 2 \times 10^5, 2 \times 10^7 \, \mu M^{-1} s^{-1}$ |
| b1, b2, b3, b4, b5 | $52, 0.21, 377.2, 0.029, 1.64 \, s^{-1}$ |
| kFlux_IP3R | $1.19 \times 10^8 \, M^{-1} s^{-1}$ |
| SERCA pump | Higgins et al.[33] |
| X0Y0, Y0X0 | $0.022$, $7.2 \, s^{-1}$ |
| X2Y2, Y2X2 | $10.8$, $75.08 \, s^{-1}$ |
| X10, Y10 | $83.67$, $30.012 \, s^{-1}$ |
| X12, Y12 | $1 \times 10^8, 1 \times 10^5 \, M^{-1} s^{-1}$ |
| PMCA pump | Penheiter et al.[75], Brini and Carafoli[76] |
| Association rate (kpm1) | $1.5 \times 10^8 \, M^{-1} s^{-1}$ |
| Dissociation rate (kpm2) | $20 \, s^{-1}$ |
| Transition rates (kpm3, kpm4) | $100$, $1.0 \times 10^5 \, s^{-1}$ |
| Leakrate (kpmleak) | $12.5 \, s^{-1}$ |
| VDCC | Bischofberger et al.[77] |
| a10, a20, a30, a40 | $4.04, 6.70, 4.39, 17.33 \, ms^{-1}$ |
| b10, b20, b30, b40 | $2.88, 6.30, 8.16, 1.84 \, ms^{-1}$ |
| V1, V2, V3, V4 | $49.14, 42.08, 55.31, 26.55 \, mV$ |
| KFlux_VDCC(V) | $\frac{AVN_A[B - e^{-V/C}]}{2f[1 - e^{V/C}]}$ |
|  | $A = 3.72 \, pS, B = 0.3933, C = 80.36 \, mV$ |
| Calbindin-D28k | Nägerl et al.[80] |
| kh+, km+ | $0.55 \times 10^7, 4.35 \times 10^7 \, M^{-1} s^{-1}$ |
| kh−, km− | $2.6$, $35.8 \, s^{-1}$ |
| Calcium sensor model | Nadkarni et al.[24] |
| Association rates (ks+, ka+) | $0.612 \times 10^8, 3.82 \times 10^6 \, M^{-1} s^{-1}$ |
| Dissociation rates (ks−, ka−) | $2.32 \times 10^3, 13 \, s^{-1}$ |
| a, b | $0.025$, $0.25$ |
| $\gamma, \delta, \epsilon$ | $2 \times 10^3 \, s^{-1}, 0.417 \times 10^{-3} \, s^{-1}, 6.34 \, ms$ |

components are provided in Fig. 7 and Table 1, respectively. We have assumed the system to be at normal body temperature (34 ºC).

*Ryanodine receptor.* The model incorporates the low and high activity of ryanodine receptors as described by Saftenku et al.[73] for RyR receptors. $O1_L, O2_L, O3_L, O1_H, O2_H$ are the open states of ryanodine receptors in response to calcium binding (see Fig. 7a1–3).

*IP3 receptor.* The De Young and Keizer[74] model was used to implement the IP3 receptor (see Fig. 7b1–3). IP3 receptor has three binding sites represented by the subscripts $i$, $j$, and $k$ in $S_{ijk}$. IP3 bound to $i$-binding site is represented as $i = 1$ and $j = 1$ represents $Ca^{2+}$ bound to activating site $j$, while $k = 1$ represents $Ca^{2+}$ bound to receptor inhibiting site $k$. Consequently, $S_{110}$ is the open IP3 receptor state.

*SERCA pumps.* We have used the four-state model for the SERCA pump described by Higgins et al.[33]. To implement it in MCell, each $Ca^{2+}$-binding step was explicitly incorporated, making it a six-state model (see Fig. 7c). Rate constants X0Y0, Y0X0,

X2Y2, and Y2X2 were modified from the original values to obtain the ER calcium-refilling rate constant of ~10 s and maintain a steady $[Ca^{2+}]_{ER}$ at ~250 μM[57,58]. SERCA surface density of 5500 μm$^{-2}$ was calculated based on Higgins et al.[33].

*PMCA pump.* The kinetic model for the PMCA pump has been taken from Penheiter et al.[75,76] and incorporates calcium leak across the plasma membrane (see Fig. 7d).

*VDCC.* Our model describes the activity of high-threshold Cav2.1, Cav2.2, and Cav2.3 type calcium channels that are activated by action potential and mediate vesicle release in a CA3 presynaptic terminal. It is based on Cav2.1 (P/Q-type) as characterized by Bischofberger et al.[77]. The microscopic rate constants $\alpha_i$ and $\beta_i$ ($i = 1, 2, 3, 4$) are exponentially dependent on voltage as $a_i(V) = a_{i0}\exp(V/V_i)$ and $b_i(V) = b_{i0}\exp(-V/Vi)$ (see Fig. 7e). The extracellular calcium concentration in our model is 2 mM and is consistent with the experimental data[26,78,79].

*Calbindin-D28k.* Calbindin-D28k was modeled according to Nägerl et al.[80]. This model contains two high-affinity sites and two medium affinity sites (see Fig. 7f).

*Calcium sensors.* The model for calcium sensor for vesicle release at CA3 terminal has been from our previous work[24]. This model incorporates calcium-dependent fast synchronous and slow asynchronous, and calcium-independent spontaneous modes of vesicle release (see Fig. 7g). Rates for synchronous, asynchronous, and spontaneous release, $\gamma$, $a\gamma$, and $\delta$, respectively were matched to experimental data. The refractory period was implemented with a time constant, $\epsilon$ (see Table 1).

**Model configurations.** Model is set up to have $Ca^{2+}$ concentration in the cytosol at 100 nM and 250 μM in ER when there is no activity and 2 mM extracellular concentration. For frequency facilitation, 20 APs at 10, 20, and 50 Hz were simulated. For paired-pulse ratio, ISI was varied from 20 to 200 ms with an interval of 10 ms until 100 ms and then with a step of 20 ms until 200 ms. To vary the vesicle release probability, calcium influx through VDCCs was varied by changing the number of VDCCs from 40 to 160. Calcium diffusion profile is not modified with calbindin-D28k mobility (see Supplementary Fig. 12). We have therefore set calbindin diffusion to zero without loss of accuracy for computational efficiency. Initial RRP size has been kept at 7 throughout all the simulations unless otherwise mentioned. The activity of SERCA pumps in response to calcium influx on the ER membrane along with PMCA pumps on the cytoplasmic membrane maintains three distinct pools of calcium (resting calcium concentration of ~250 μM in the ER and 100 nM in the cytosol and 2 mM extracellular space).

**Simulations.** All simulations had a time-step of 1 μs. In simulations for paired-pulse stimulus, 5000 trials were used to calculate average response for VDCC = 40–60, 2000 trials were used for VDCC = 70–90, and 1000 trials for VDCC = 100–160. Release probability in response to an AP is calculated by counting the number of trials that resulted in vesicular release within 20 ms after initiation of AP, divided by the total number of trials. This is in accordance with the definition of vesicle release probability: the probability that at least one vesicle is released. Error in release probability was calculated using 1000 resampling with replacement and then mean and standard error was calculated from the resampled data. Simulations were run for varied times depending on the stimulus protocols used. For the paired-pulse protocol, the total simulation time was 30 ms + ISI. For train stimulus of 10 Hz (Supplementary Fig. 6), 20 Hz (Fig. 4), and 50 Hz (Supplementary Fig. 7), total simulation time was 1.050 s, 2.050 s, and 450 ms, respectively. On average, simulating a single trial of 50 ms requires about 20 min. A description of all the fitting functions used in the figures is provided in Supplementary Note 1.

**Statistics and reproducibility.** For each case, 2000 trails were simulated. Analyses were performed using the python programming language. All the python scripts are available along with the model code. Plots were generated using python programming language and gnuplot. Data expressed as mean ± s.d. as indicated.

**Reporting summary.** Further information on research design is available in the Nature Research Reporting Summary linked to this article.

## Data availability
Data available on request from the authors.

## Code availability
The complete model code and analysis script can be accessed at https://github.com/CNLiiserp/CA3bouton.

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

## Acknowledgements

This work was supported by the Wellcome Trust/DBT India Alliance Grant IA/I/12/1/500529, the DST/INSPIRE Fellowship /2015/IF150919 and the Indian Institute of Science Education and Research Pune.

## Author contributions

N.S.: conceptualization, methodology, software, validation, formal analysis, investigation, data curation, writing original draft, visualization, and funding acquisition. T.B.: conceptualization, methodology, software, formal analysis, investigation, writing review and editing, and visualization. H.L.: conceptualization, resources, writing review and editing, and funding acquisition. T.S.: conceptualization, resources, writing review and editing, and funding acquisition. S.N.: conceptualization, methodology, software, validation, formal analysis, resources, investigation, writing original draft, visualization, supervision, project administration, and funding acquisition.

## Competing interests

The authors declare no competing interests.
