## [Peer Review File · Communications Biology]

Reviewers' comments:

Reviewer #1 (Remarks to the Author):

Please see attached file for formatted version

Review for

Presynaptic endoplasmic reticulum regulates short-term plasticity in hippocampal synapses

Nishant Singh¹, Thomas Bartol², Herbert Levine³, Terrence Sejnowski², and Suhita Nadkarni^{1,*}

First, let me say that I thoroughly enjoyed reading this paper. It was very carefully done, with great attention to detail. The results are compelling and have broad application in other synapses, and other processes that use calcium as a second messenger molecule. I believe it should be published more or less as is, and I have only a few comments and corrections.

What are the major claims of the paper? Are they novel and will they be of interest to others in the community and the wider field?

The authors use an "in silico" approach, with M-Cell, to simulate the effect of certain properties of the ER on short term plasticity at the CA1-CA3 synapse. ER is present in the Schaffer collaterals, but so far its contribution to signaling properties at these synapses has not been studied. It is therefore novel, and because abnormal STP is an indicator of dementia, and, indeed, Alzheimer's disease (AD), it will be of interest to the community and to a wider audience. Most interesting is the connection between ER function blocking and compromised plasticity in AD, seen in experiments (Zhang et al. 2009, 2010).

M-Cell is used to good advantage here, as direct measurements of local calcium in the active zone in this small synapse are not yet possible.

With a sequence of experiments using PPR as a measure of STP, the authors are able to establish the role of presynaptic ER in STP, and that its effect on STP occurs primarily through the buffering of the second messenger molecule, calcium, by SERCA pumps on the ER membrane. This buffering creates a lower Pr at the synapse when it receives the first pulse, which in turn causes larger response upon a second pulse (facilitation) through preservation of the readily releasable pool of vesicles, and by retaining calcium near the active zone. They established this by comparing synapses with and without the SERCA pumps blocked.

In one set of simulations they broke down the response of synapses responding to single APs by showing time traces of calcium flux through RyR, IP3R and VDCC compared to the action potential itself, clarifying the relative time scales of each. Similarly, they showed the number of Ca ions bound to SERCA and calbindin, and free calcium in the AZ and cytosol. Open RyR and IP3R channels vs time were plotted, and the transfer of Ca via PMCA. Finally, stacked histograms representing number of synchronous and asynchronous release are shown. Figure 2d shows an initial rapid reduction of Ca, followed by a linear trend upward. Is this related to the plot in 2e, which shows the fast opening RyR and the longer lived, slower opening IP3R? I was not able to find any reference to this interesting "boundary layer" effect in the body of the paper, but perhaps I missed it.

Next they present results that show how blocking the SERCA pump effects STP. The synapse being studied is a low Pr terminal, and should have a large PPR. By running side by side simulations of paired pulse protocols with a control synapse and one with the SERCA pumps blocked, they could study the effect of the blocked pump on the PPR. The basic effect of the pump is to buffer calcium released upon the first stimulus, which then slowly unbinds, keeping the baseline concentration of calcium higher than before. The second stimulus then can be larger than the first, causing facilitation, e.g. a large PPR. They found that blocking SERCA pumps substantially diminished the PPR, at all Pr,

but especially at the low Pr synapses like the CA3-CA1.

Figure 3 shows the effect of blocking SERCA pumps on experiments with a stimulus train. The conclusions from the PPR hold in this scenario as well, but are complicated by the action of IP3R and RYR. The explanation of this is a little garbled (line 141), and could be tidied up.

The case then is made for increased reliability through increased presynaptic release probability by the machinery of calcium buffering in the synapse. They find that the coefficient of variation (CV) of the second response (Pr2) decreases with Pr1, and that the CV in the control case is lower than in the SERCA blocked case. Then they study 'synapse-specific' history as opposed to pooling the results from many independent synapses, using the probability that a successful transmission event in response to a first stimulus is followed by another successful transmission event in response the second stimulus, P_{11} . Because P_{11} is higher for synapses with calcium stores, they claim these synapses retain a better history of release through calcium dynamics. As efferent synapses are involved in critical computations and short term plasticity enables estimating non-discrete signals arriving at the synapse, they make some claims, that are somewhat vague, concerning release rate coding at these synapse. The mechanism for larger P_{11} in synapses with calcium stores is then described (lines 180-196), which, because it is complicated, is hard to follow. Perhaps a flow chart could be used for this, and for descriptions of other more complicated mechanisms, involving several competing factors (e.g. lines 91-111 and lines 137-144). While this is probably not standard practice in these types of papers, it certainly would aid understanding and thus make the arguments more convincing.

Is the work convincing, and if not, what further evidence would be required to strengthen the conclusions? On a more subjective note, do you feel that the paper will influence thinking in the field? Please feel free to raise any further questions and concerns about the paper.

The work was made more convincing by the comparison with experiments by Dobrunz (1997) and Murthy (1997). PPR vs intrinsic release probability were plotted for 40 and 50 ms ISIs from both authors, and the simulations, as well as PPR vs ISI, and PPR vs readily releasable pool size. It was found that the simulations of the control case matched the experiments very tightly, while the SERCA blocked simulations did not. That said, the use of eq (1) was not well developed. What are the variables? Which set of data was it applied to? Seemingly to the data in figure 6a). What were the fit values of a and b? Maybe the details can put in as supplemental material, or the equation could be left out entirely. Also, where did the data in figure 6 d) come from? How is it compared with experiments? Through the value of tau? This could be made a little clearer in the body of the paper. In the Discussion the authors summarize their results and justify some of their modeling choices, with back-up evidence presented in the Supplementary Material (e.g. robustness of results with respect to exact placement of the various channels). Furthermore, the use of "in-silico" experiments is justified in light of the difficulty of measuring calcium in the active zone directly in these synapses. They finish by discussing the fact that abnormal STP is one of the earliest indicators of dementia, where modified facilitation is seen specifically in AD synapses in an animal model. These experiments also indicated that disrupted ER signaling underlies the compromised plasticity in AD, giving this study's findings increased significance and relevance.

We would also be grateful if you could comment on the appropriateness and validity of any statistical analysis, as well the ability of a researcher to reproduce the work, given the level of detail provided.

The detail given in the methods section should be enough for another researcher to reproduce the work. It was carefully and thoroughly described.

Specific comments on text:

First line in abstract is a little awkward.

Line 55: remove "majorly"

Line 59: Add comma: Also, the local calcium concentration despite dropping rapidly, does

Line 74: extensive reports ON presynaptic RyRs

Line 107: contributes to facilitate vesicle release to subsequent stimulus is awkward.
Change to "facilitates vesicle release upon subsequent stimulation".

Line 116: awkward sentence, replacement "Does not contribute to the calcium signal in a significant way"

Line 137: remove "This is because of the". The transition from the previous paragraph is not clear.

Line 138: Ryanodine receptors, owing to a fast, albeit low, affinity calcium binding site, can get be activated by fast and high amplitudes of calcium.

Line 140-41: Confusing construction.

Line 144: where are these figures?

Line 176-178: confusing constructed sentence

Line 196: what is a synaptic memory trace?

Line 235: cause a lower Pr. The Inverse relationship between

Line 237: As binding sites of SERCA get become occupied with upon subsequent pulses, the buffering capacity of SERCAs....

Line 250: would steer the synapse towards much quicker depletion.... is awkward

Line 276: why is PPR a "high-dimensional" measure of plasticity?

Reviewer #2 (Remarks to the Author):

This paper investigates the role of presynaptic endoplasmic reticulum (ER) in synaptic short term plasticity. Specifically, the mechanism considered here is ER-dependent sequestering and release of calcium ions, and the focus of this paper is on the effects this has on presynaptic calcium concentrations at vesicle release sites. To investigate this, the authors use Monte Carlo simulations of single molecule diffusion and stochastic molecular reactions. Simulations were carried out both using a realistic synaptic morphology (an excitatory synapse in CA3) from an EM reconstruction, and with simplified morphologies to confirm central results. Overall I find this a nice and interesting paper that sheds further light on the intricacies of the synaptic release process. It is perhaps most intriguing to see how active zone Ca^{++} , various calcium stores and vesicle release interact as this is indeed highly non-trivial.

A main question I have about this work is whether the effect of ER stores is fully explainable by the effect these have on the release probability (Pr), and the consequent effect on vesicle depletion. This seems plausible to me, and in this case I expect that the paired pulse experiments would be explainable as follows: the first release is smaller solely because Pr is lower, and the second release is enhanced because a somewhat higher residual Ca^{++} adds to the Ca^{++} transient, increasing Pr correspondingly. Subsequent action potentials then eventually cause weaker release as the RRVP gets depleted. So overall fewer vesicles are released during a train of action potentials when the ER buffering is active (compared to the "stores blocked condition"). Is this correct? If so, how does this compare to the effect of just reducing Pr (fewer VDCCs)? Can this (a lower number of VDCCs when stores are blocked) give rise to essentially the same behaviour, or does the increased residual Ca^{++} contribute something else?

In figures 3 and 4 it would therefore be interesting to also see the the actual number of released vesicles, which would correspond to $Pr * RRVP$. Also, how does asynchronous release differ between "control" and "buffers blocked conditions"? This does not seem to lead to much additional vesicle depletion, but would it be a measurable difference in an experiment?

In lines 131 and following (until end of subsection): I did not understand this section, and the referenced panels figure 4g and 4h do not seem to exist. Do you want to say that release from ER

buffers contributes to a residual Ca⁺⁺ level that adds to the AZ Ca⁺⁺ at the next action potential? I could not entirely follow the discussion of the IP3R/RyR effects on facilitation.

In the section on increased reliability, (beginning line 145): I'm not sure about the definition of the CV in relation to the release probability. In my opinion the factor n in the equation should be the number of vesicles in the RRVP, and the CV then approximates the variability seen when comparing the standard deviation of the number of released vesicles to the mean (over trials). This can be checked by computing $\sqrt{np(1-p)}/np$, where p is the release probability and n the RRVP size, and compared to the CV estimated from multiple simulations.

Now, the main question here is if the difference in CV of Pr2 (Fig 5a) is correctly predicted by the corresponding differences in release probability and vesicle depletion? This could be checked directly (and the lines in the plot could show the prediction). The same could be done for the conditional probability P11 (two successful transmission events, Fig 5b). Note here I expect a lower probability for successful transmission to the first input, is this correct? If so, it may be fairer to show this number as well as it does not mean the synapse is more reliable in all cases. Finally, I am not quite convinced the variability of the Ca⁺⁺ concentration matters here. It is certainly lower between action potentials (which is interesting), but not when Ca⁺⁺ channels open. So it looks to me this does not really affect the reliability of the synapse, unless I missed something.

So overall I think the effects shown here may have simpler explanations than the paper suggests (I may be wrong, of course). I think it is worth checking the results against a binomial release model to see where this correctly predicts the effects, and where there are discrepancies. I think this would not require any new simulations, and would help to understand the results in the context of simpler, but more tractable release models.

Minor comments:

In all figures there are lines that approximate the data points - how were these made? In Figure 5b, the blue line looks like a quadratic function, and while connecting the points, decreases for high Pr1 without support from a data point. Is this intended? While the lines help guide the eye, they may be misleading, so please at least state how they were drawn.

line 53: ...Consequently, rapid increase and fall in calcium concentration *is* closely...

line 61: Perhaps define PMCA pump at first mention.

line 131: *The* control synapse has lower peak calcium...

line 156: The CV is usually called the Coefficient of Variation, and should be written as: $CV = \sqrt{(1-Pr2)/(n*PR)}$

The resulting model is thus complex and has many parameters

line 410: simulation time step seems to have wrong units (1s)

Reviewer #1 (Remarks to the Author):

First, let me say that I thoroughly enjoyed reading this paper. It was very carefully done, with great attention to detail. The results are compelling and have broad application in other synapses, and other processes that use calcium as a second messenger molecule. I believe it should be published more or less as is, and I have only a few comments and corrections. What are the major claims of the paper? Are they novel and will they be of interest to others in the community and the wider field? The authors use an “in silico” approach, with M-Cell, to simulate the effect of certain properties of the ER on short term plasticity at the CA1-CA3 synapse. ER is present in the Schaffer collaterals, but so far its contribution to signaling properties at these synapses has not been studied. It is therefore novel, and because abnormal STP is an indicator of dementia, and, indeed, Alzheimer’s disease (AD), it will be of interest to the community and to a wider audience. Most interesting is the connection between ER function blocking and compromised plasticity in AD, seen in experiments (Zhang et al. 2009, 2010). M-Cell is used to good advantage here, as direct measurements of local calcium in the active zone in this small synapse are not yet possible. With a sequence of experiments using PPR as a measure of STP, the authors are able to establish the role of presynaptic ER in STP, and that its effect on STP occurs primarily through the buffering of the second messenger molecule, calcium, by SERCA pumps on the ER membrane. This buffering creates a lower Pr at the synapse when it receives the first pulse, which in turn causes larger response upon a second pulse (facilitation) through preservation of the readily releasable pool of vesicles, and by retaining calcium near the active zone. They established this by comparing synapses with and without the SERCA pumps blocked. In one set of simulations they broke down the response of synapses responding to single APs by showing time traces of calcium flux through RyR, IP3R and VDCC compared to the action potential itself, clarifying the relative time scales of each. Similarly, they showed the number of Ca ions bound to SERCA and calbindin, and free calcium in the AZ and cytosol. Open RyR and IP3R channels vs time were plotted, and the transfer of Ca via PMCA. Finally, stacked histograms representing number of synchronous and asynchronous release are shown.

We sincerely thank the reviewers for their appreciation of our work.

Figure 2d shows an initial rapid reduction of Ca, followed by a linear trend upward. Is this related to the plot in 2e, which shows the fast opening RyR and the longer lived, slower opening IP3R? I was not able to find any reference to this interesting “boundary layer” effect in the body of the paper, but perhaps I missed it.

The cumulative extrusion of Ca ions via PMCA pumps is shown in Figure 2d. We missed adding ‘cumulative’ to the label for the ‘Y’ axis here. This has been fixed in the revised version. We apologize for the mistake. The sharp rise in cytosolic Ca concentration from the opening of VDCCs (Figure 2A, Orange) is reflected in the initial rapid activity of PMCAs. At about 5 ms, VDCCs are closed and the release of Ca ions by IP3R and RyR into the cytosol from endoplasmic reticulum contributes to the later part of the action of PMCAs. References for the kinetic details of IP3 and RyR receptors are provided in methods section and supplementary figure S9.

Next, they present results that show how blocking the SERCA pump effects STP. The synapse being studied is a low Pr terminal, and should have a large PPR. By running side by side simulations of paired pulse protocols with a control synapse and one with the SERCA pumps blocked, they could study the effect of the blocked pump on the PPR. The basic effect

of the pump is to buffer calcium released upon the first stimulus, which then slowly unbinds, keeping the baseline concentration of calcium higher than before. The second stimulus then can be larger than the first, causing facilitation, e.g. a large PPR. They found that blocking SERCA pumps substantially diminished the PPR, at all Pr, but especially at the low Pr synapses like the CA3-CA1. Figure 3 shows the effect of blocking SERCA pumps on experiments with a stimulus train. The conclusions from the PPR hold in this scenario as well, but are complicated by the action of IP3R and RYR. The explanation of this is a little garbled (line 141), and could be tidied up.

This entire paragraph (previously, line 137-144) was meant to be deleted from the manuscript but inadvertently slipped into the submitted version of the manuscript. We have removed this paragraph. We apologize for the mistake. A more detailed version of this description appropriately appears in the supplementary information (see figure S9).

The case then is made for increased reliability through increased presynaptic release probability by the machinery of calcium buffering in the synapse. They find that the coefficient of variation (CV) of the second response (Pr2) decreases with Pr1, and that the CV in the control case is lower than in the SERCA blocked case. Then they study 'synapse-specific' history as opposed to pooling the results from many independent synapses, using the probability that a successful transmission event in response to a first stimulus is followed by another successful transmission event in response to the second stimulus, P₁₁. Because P₁₁ is higher for synapses with calcium stores, they claim these synapses retain a better history of release through calcium dynamics. As efferent synapses are involved in critical computations and short term plasticity enables estimating non-discrete signals arriving at the synapse, they make some claims, that are somewhat vague, concerning release rate coding at these synapses. The mechanism for larger P₁₁ in synapses with calcium stores is then described (lines 180-196), which, because it is complicated, is hard to follow. Perhaps a flow chart could be used for this, and for descriptions of other more complicated mechanisms, involving several competing factors (e.g. lines 91-111 and lines 137-144). While this is probably not standard practice in these types of papers, it certainly would aid understanding and thus make the arguments more convincing.

We have taken the feedback from this review and made major modifications to the write-up of this section ("Increased reliability mediated by store machinery"). We believe that it now reads clearly.

Is the work convincing, and if not, what further evidence would be required to strengthen the conclusions? On a more subjective note, do you feel that the paper will influence thinking in the field? Please feel free to raise any further questions and concerns about the paper.

The work was made more convincing by the comparison with experiments by Dobrunz (1997) and Murthy (1997). PPR vs intrinsic release probability were plotted for 40 and 50 ms ISIs from both authors, and the simulations, as well as PPR vs ISI, and PPR vs readily releasable pool size. It was found that the simulations of the control case matched the experiments very tightly, while the SERCA blocked simulations did not. That said, the use of eq (1) was not well developed. What are the variables? Which set of data was it applied to? Seemingly to the data in figure 6a). What were the fit values of a and b? Maybe the details can be put in as supplemental material, or the equation could be left out entirely.

The curve-fitting function we have used does not give new insight into the data. This was only meant as a visual guide, however, we agree that this can be confusing. We therefore have removed it from the manuscript body and provided a description to all the fitting functions in supplementary information.

Also, where did the data in figure 6 d) come from? How is it compared with experiments? Through the value of tau? This could be made a little clearer in the body of the paper.

The data from the 'Control' model is plotted in fig. 6d. In the paper by Dobrunz and Stevens, 1997, the vesicle replenishment timescale was measured to be ~3 sec in CA3-CA1 synapse. Our model also has a vesicle replenishment timescale of ~3.0 sec. We have rewritten this part in the manuscript. We apologize that this was not clearly stated in the previous version.

In the Discussion the authors summarize their results and justify some of their modeling choices, with back-up evidence presented in the Supplementary Material (e.g. robustness of results with respect to exact placement of the various channels). Furthermore, the use of "in-silico" experiments is justified in light of the difficulty of measuring calcium in the active zone directly in these synapses. They finish by discussing the fact that abnormal STP is one of the earliest indicators of dementia, where modified facilitation is seen specifically in AD synapses in an animal model. These experiments also indicated that disrupted ER signaling underlies the compromised plasticity in AD, giving this study's findings increased significance and relevance. We would also be grateful if you could comment on the appropriateness and validity of any statistical analysis, as well the ability of a researcher to reproduce the work, given the level of detail provided. The detail given in the methods section should be enough for another researcher to reproduce the work. It was carefully and thoroughly described.

Specific comments on text:

First line in abstract is a little awkward.

We have made the following change: "Short-term plasticity preserves a brief history of synaptic activity that is communicated to the postsynaptic neuron."

Line 55: remove "majorly"

"Majorly" has been removed.

Line 59: Add comma: Also, the local calcium concentration despite dropping rapidly, does

Comma has been added.

Line 74: extensive reports ON presynaptic RyRs

Change has been made.

Line 107: contributes to facilitate vesicle release to subsequent stimulus is awkward.

Change to "facilitates vesicle release upon subsequent stimulation".

Change has been made.

Line 116: awkward sentence, replacement "Does not contribute to the calcium signal in a significant way"

The sentence has been changed to “Does not contribute to the calcium signal significantly”.

Line 137: remove “This is because of the”. The transition from the previous paragraph is not clear.

Line 138: Ryanodine receptors, owing to a fast, albeit low, affinity calcium binding 139 site, can get be activated by fast and high amplitudes of calcium.

Line 140-41: Confusing construction.

Line 144: where are these figures?

The entire paragraph (previously, line 137-144) was meant to be deleted from the manuscript but inadvertently slipped into the submitted version. We apologize for the mistake. A more detailed version of this description appropriately appears in the supplementary information (see figure S9).

Line 176-178: confusing constructed sentence

The entire section has been modified. We believe that the explanation is clearer now.

Line 196: what is a synaptic memory trace?

Thank you for this feedback. We realize that ‘synaptic memory trace’ is ambiguous. We have modified the sentence to “In summary, intracellular calcium stores allow synapses to operate at low intrinsic release probability and yet exhibit large, facilitated release with a lower CV caused by longer-lasting and less noisy residual calcium.” [Line no. 198-200]

Line 235: cause a lower Pr. The Inverse relationship between

The sentence has been changed to “SERCA’s calcium buffering properties, in turn, lowers Pr.”

Line 237: As binding sites of SERCA get become occupied with upon subsequent pulses, the buffering capacity of SERCAs....

The sentence has been changed to “Calcium from consecutive pulses binds to the open SERCA sites. This decrease in the overall availability of SERCA also decreases its buffering capacity ...” [Line no. 248-250]

Line 250: would steer the synapse towards much quicker depletion.... is awkward

The sentence has been changed to “... but at high Pr, the small RRP size at this synapse would deplete quickly”. [Line no. 261]

Line 276: why is PPR a “high-dimensional” measure of plasticity?

By high-dimensional measure, we meant that several intrinsic and interdependent variables of the synapse can modulate the value of PPR. It depends on — calcium concentration, number of VDCCs, RRP size, Intrinsic Pr, coupling distance of VDCC and active zone, SERCA, buffers, and synaptotagmin activity. [Line no. 285-287]

Reviewer #2 (Remarks to the Author):

This paper investigates the role of presynaptic endoplasmic reticulum (ER) in synaptic short term plasticity. Specifically, the mechanism considered here is ER-dependent sequestering and release of calcium ions, and the focus of this paper is on the effects this has on presynaptic calcium concentrations at vesicle release sites. To investigate this, the authors use Monte Carlo simulations of single molecule diffusion and stochastic molecular reactions. Simulations were carried out both using a realistic synaptic morphology (an excitatory synapse in CA3) from an EM reconstruction, and with simplified morphologies to confirm central results. Overall I find this a nice and interesting paper that sheds further light on the intricacies of the synaptic release process. It is perhaps most intriguing to see how active zone Ca^{++} , various calcium stores and vesicle release interact as this is indeed highly non-trivial.

A main question I have about this work is whether the effect of ER stores is fully explainable by the effect these have on the release probability (Pr), and the consequent effect on vesicle depletion. This seems plausible to me, and in this case I expect that the paired pulse experiments would be explainable as follows: the first release is smaller solely because Pr is lower, and the second release is enhanced because a somewhat higher residual Ca^{++} adds to the Ca^{++} transient, increasing Pr correspondingly. Subsequent action potentials then eventually cause weaker release as the RRVP gets depleted. So overall fewer vesicles are released during a train of action potentials when the ER buffering is active (compared to the "stores blocked condition"). Is this correct?

Increased facilitation in synapses with ER is indeed mediated by two mechanisms: 1) Smaller first release (Pr) due to SERCA buffering leads to bigger increases in subsequent releases. 2) Increases in ensuing releases also result from an increase in basal calcium after each pulse as a consequence of calcium release from the stores and higher occupancy of SERCA (lower buffering capacity). This mechanism, as the reviewer correctly describes, conserves a limited vesicle pool while facilitating release. This subsequent increase lasts till the effect of RRP depletion takes over for both the synaptic configurations (as you may see in fig. 4a,b and c).

If so, how does this compare to the effect of just reducing Pr (fewer VDCCs)? Can this (a lower number of VDCCs when stores are blocked) give rise to essentially the same behaviour, or does the increased residual Ca^{++} contribute something else?

The reduction in the number of VDCCs causes a decrease in Pr. Consistent with several classical studies on CA3-CA1 synapses (Murthy et al., 1997, Neuron; Zhang et al., 2009, Nature), PPR in both the configurations (Control and Stores blocked) is inversely related Pr. However, for the same Pr, we have shown that the facilitation is enhanced in the 'Control' configuration for various stimulus protocols. Separately, for the same VDCCs, 'Control' configuration again shows enhanced facilitation compared to 'Stores-blocked'. This is because of the enhanced basal level of calcium and lower intrinsic Pr for 'Control' configuration (see figure 3c2).

In figures 3 and 4 it would therefore be interesting to also see the the actual number of released vesicles, which would correspond to $Pr * RRVP$. Also, how does asynchronous release differ between "control" and "buffers blocked conditions"? This does not seem to lead to much additional vesicle depletion, but would it be a measurable difference in an experiment?

We would like to thank the reviewer for this suggestion. We now have additional supplemental figures (fig. S2 and S8) showing differences in asynchronous releases as well as total releases. In Control synapses, we observe consistently higher asynchronous release compared to stores blocked synapses. This is observed in both paired-pulse and train stimulus. These differences are further accentuated over longer stimulus protocol (fig. S8). This leads us to believe that any experimental data set of prolonged stimulus over multiple trials will show a measurable difference.

In lines 131 and following (until end of subsection): I did not understand this section, and the referenced panels figure 4g and 4h do not seem to exist. Do you want to say that release from ER buffers contributes to a residual Ca^{++} level that adds to the AZ Ca^{++} at the next action potential? I could not entirely follow the discussion of the IP3R/RyR effects on facilitation.

This entire paragraph (previously, line 137-144) was meant to be deleted from the manuscript but inadvertently slipped into the submitted version of the manuscript. We have removed this paragraph. We apologize for the mistake. A more detailed version of this description appropriately appears in the supplementary information (see figure S9).

In the section on increased reliability, (beginning line 145): I'm not sure about the definition of the CV in relation to the release probability. In my opinion the factor n in the equation should be the number of vesicles in the RRVP, and the CV then approximates the variability seen when comparing the standard deviation of the number of released vesicles to the mean (over trials). This can be checked by computing $\sqrt{np(1-p)}/np$, where p is the release probability and n the RRVP size, and compared to the CV estimated from multiple simulations.

We would like to thank the reviewer for a careful evaluation of our manuscript. We made a mistake in the statement of equation for CV ($= \sqrt{(1-p)/(np)}$), p is the release probability and n is the number of active zone/release sites); we have now corrected it. As per the recommendation of the reviewer, to compare with quantal theory, we measured CV of Pr_1 of our data and compared it with predicted values (see supplementary fig. S10).

Now, the main question here is if the difference in CV of Pr_2 (Fig 5a) is correctly predicted by the corresponding differences in release probability and vesicle depletion? This could be checked directly (and the lines in the plot could show the prediction). The same could be done for the conditional probability P_{11} (two successful transmission events, Fig 5b) . Note here I expect a lower probability for successful transmission to the first input, is this correct? If so, it may be fairer to show this number as well as it does not mean the synapse is more reliable in all cases.

The reviewer correctly points out that P_{11} is influenced by depletion. Apart from depletion, both P_{11} and Pr_2 are also influenced by facilitation. As we have described in our manuscript, facilitation in our model is mediated by multiple interconnected processes. Describing it by differences in release probability and vesicle depletion would therefore not be appropriate for this data. Facilitation for P_{11} takes place because calcium signal causally linked to the successful first release, Pr_1 corresponds to a magnitude that is 'bigger' than average. This

leads to an enhanced probability of release to the subsequent signal. As shown below (and in the manuscript), P_{11} is also a function of Pr_1 (black line describes synapses with no facilitation case). However, the crucial result we want to describe is the difference in P_{11} for the two types of synaptic configurations (Control and without stores). In fig. 5b and c, we show that for the same Pr_1 , P_{11} is lower in stores blocked case compared to Control. This leads us to make the ‘reliability’ statement for the synapse with stores (Control); success of release in the first pulse better predicts success of release of the second in the Control synapse.

Finally, I am not quite convinced the variability of the Ca^{++} concentration matters here. It is certainly lower between action potentials (which is interesting), but not when Ca^{++} channels open. So it looks to me this does not really affect the reliability of the synapse, unless I missed something. So overall I think the effects shown here may have simpler explanations than the paper suggests (I may be wrong, of course). I think it is worth checking the results against a binomial release model to see where this correctly predicts the effects, and where there are discrepancies. I think this would not require any new simulations, and would help to understand the results in the context of simpler, but more tractable release models.

The set of figures 5d-h describe the mechanism for the observed enhanced P_{11} in Control synapses. As we mentioned before, the calcium signal associated with a successful release that is followed by another successful release, is larger than average. This is the basis of facilitation seen in P_{11} . A larger number of calcium channels are needed for the same release probability in Control synapses as compared to Stores blocked (described in detail in the result section “Blocking ER compromises short-term plasticity”). This leads to lower variability of calcium flux through the voltage gated calcium channels (figure 5d). This essentially means calcium signal is larger consistently over multiple trials in synapses with stores and therefore enhances the success of the second release.

Yet another mechanism for lower variability of calcium in synapses with stores is the presence of SERCA pumps. Role of SERCA in lowering the variability of calcium is described in figure 5e-g. The isolated influence of SERCA without the contribution of voltage gated calcium channels in lowering the variability of calcium is described in Figure 5h. Essentially, we simulated synapses with an identical number of VDCCs for both Control and Stores-blocked. The effect of SERCA is clearly seen in between the pulses wherein the CV of calcium signal at the AZ remains lower for Control synapses. The reviewer is correct in pointing out that this alone cannot explain the enhanced P_{11} seen in Control, but adds to the effect of a larger number of VDCCs associated with Control. Needless to say, CV of calcium signal underlying P_{11} for the synapse with stores will be lower throughout the stimulus compared to synapse of the Stores blocked case with the same Pr. The lower variability of calcium signal associated with the Control is also an independent interesting observation of our model given the various downstream signaling implications of calcium over a wide range

of time scales (for example, vesicle recycling). We therefore thought it was important to report it (since this would also be difficult to measure experimentally). We now clarify it in this section. We appreciate the feedback from the referee and have rewritten this section.

Minor comments:

In all figures there are lines that approximate the data points - how were these made? In Figure 5b, the blue line looks like a quadratic function, and while connecting the points, decreases for high Pr1 without support from a data point. Is this intended? While the lines help guide the eye, they may be misleading, so please at least state how they were drawn.

The curve-fitting was done mainly to serve as a visual guide and does not indicate any functional significance. We agree with the reviewer that in some places (fig. 5b) it is unnecessary and can be misleading. Description of all the fitting functions used in the figures is now provided in supplementary information.

line 53: ...Consequently, rapid increase and fall in calcium concentration **is** closely...

I doubt **is** is required in the following sentence: "Consequently, rapid increase and fall in calcium concentration closely follows the VDCC calcium flux, which, in turn, follows the voltage profile".

line 61: Perhaps define PMCA pump at first mention.

We have now defined PMCA where it first appears in the manuscript.

line 131: **The** control synapse has lower peak calcium...

"The" has been added.

line 156: The CV is usually called the Coefficient of Variation, and should be written as:
 $CV = ((1 - Pr2) / (n * PR))^{1/2}$

We would like to thank the reviewer for pointing this out. The expression has been corrected.

line 410: simulation time step seems to have wrong units (1s)

We apologize for this mistake. It has now been fixed.

REVIEWERS' COMMENTS:

Reviewer #1 (Remarks to the Author):

The author's have satisfactorily addressed all the points in my review of the manuscript.

Reviewer #2 (Remarks to the Author):

All my comments have been addressed, I have no reservations recommending piublikation.